# Identification of Toxicity Parameters Associated with Combustion Produced Soot Surface Chemistry and Particle Structure by in Vitro Assays

**DOI:** 10.3390/biomedicines8090345

**Published:** 2020-09-11

**Authors:** Heba Al Housseiny, Madhu Singh, Shaneeka Emile, Marvin Nicoleau, Randy L. Vander Wal, Patricia Silveyra

**Affiliations:** 1Biobehavioral Laboratory, School of Nursing, The University of North Carolina at Chapel Hill, Chapel Hill, NC 27599, USA; hebaal@email.unc.edu; 2John and Willie Leone Family Department of Energy and Mineral Engineering, The Pennsylvania State University, University Park, PA 16801, USA; madhusingh@alumni.psu.edu (M.S.); ruv12@psu.edu (R.L.V.W.); 3The Pennsylvania State University College of Medicine, Hershey, PA 17033, USA; snemile77@gmail.com; 4School of Medicine, Case Western Reserve University, Cleveland, OH 44106, USA; mnicoelea@fandm.edu; 5EMS Energy Institute, The Pennsylvania State University, University Park, PA 16801, USA

**Keywords:** air pollution, soot, particulate matter, lung inflammation, functional groups

## Abstract

Air pollution has become the world’s single biggest environmental health risk of the past decade, causing millions of yearly deaths worldwide. One of the dominant air pollutants is fine particulate matter (PM_2.5_), which is a product of combustion. Exposure to PM_2.5_ has been associated with decreased lung function, impaired immunity, and exacerbations of lung disease. Accumulating evidence suggests that many of the adverse health effects of PM_2.5_ exposure are associated with lung inflammation and oxidative stress. While the physical structure and surface chemistry of PM_2.5_ are surrogate measures of particle oxidative potential, little is known about their contributions to negative health effects. In this study, we used functionalized carbon black particles as surrogates for atmospherically aged combustion-formed soot to assess the effects of PM_2.5_ surface chemistry in lung cells. We exposed the BEAS-2B lung epithelial cell line to different soot at a range of concentrations and assessed cell viability, inflammation, and oxidative stress. Our results indicate that exposure to soot with varying particle surface composition results in differential cell viability rates, the expression of pro-inflammatory and oxidative stress genes, and protein carbonylation. We conclude that particle surface chemistry, specifically oxygen content, in soot modulates lung cell inflammatory and oxidative stress responses.

## 1. Introduction

Air pollution has become one of the greatest environmental health hazards to millions around the world and is primarily caused by years of industrialization and population growth, particularly in developing countries [1,2]. In the past decade, air pollution alone has been linked to 7 million annual deaths worldwide. The Global Burden of Disease Study identified air pollution as a risk factor for cardiovascular disease and respiratory infections, and it is estimated to have contributed to nearly 5 million premature deaths worldwide in 2017 alone [3,4].

Air pollutants are separated into two categories depending on the source of production. Primary pollutants such as heavy petroleum products (soot) and oxides of nitrogen (NOx) and sulfur (SOx) are emitted into the air by the combustion of fossil fuels, vehicle exhaust, natural fires, industrial practices, and natural dust [5]. Secondary pollutants are formed when primary pollutants react with other molecules in the atmosphere, altering their toxin absorption ability. Secondary pollutants include ozone, acid rain, and particulate air pollution [5].

Particulate matter (PM) is a complex mixture of volatile organic and inorganic compounds, including solid particles and liquid droplets, with different physicochemical properties and toxicity [6,7]. Carbon black, produced by an incomplete combustion of soot, comprises a significant portion of PM. The toxicity of PM is characterized by its size, surface chemistry, solubility, and ability to form reactive oxygen species. Fine PM size ranges from 2.5 μm to 10 μm in aerodynamic diameter (PM_2.5_ to PM_10_). Notably, PM_2.5_ is known to exacerbate many pre-existing heart and respiratory diseases that increase the risk of death [8]. The finest particles, PM_2.5_, have been reported to induce inflammation and oxidative stress in the respiratory tract, and they exacerbate respiratory and cardiovascular diseases including long-term chronic diseases such as cancer and asthma. Yet, the mechanisms underlying PM_2.5_-induced inflammation are still unknown [8,9,10,11,12]. The mortality rate due to PM_2.5_ is more prominent in the elderly, pregnant women, infants, and low-income populations [13,14]. Exposure to fine particles affects lower-income populations in which a significant proportion have an income below the poverty threshold, leading to environmental inequality and propagating economic burden [15,16].

Exposure of the nasal and lung epithelia to PM_2.5_ triggers an inflammatory response within the respiratory system that, if not resolved, can contribute to chronic inflammation. This inflammatory response is characterized by the production of extracellular signaling proteins, including cytokines and chemokines secreted by epithelial and immune cells, and it plays a critical role in the lung’s defense mechanism against oxidative stress [17]. Previous studies have described an increase in the expression of cellular pro-inflammatory cytokines such as interleukin 6 (IL-6) and interleukin 1 beta (IL-1β) following exposure to PM_2.5_ [18,19,20]. Protein carbonylation has also been reported as a biomarker of oxidative stress-induced protein damage and has been associated with the development of cardiovascular and respiratory diseases [21,22]. A few studies have shown that protein carbonylation is increased in PM_2.5_-treated human keratinocytes and rat epithelial lung cells [23]. Thus, while cytokine expression and protein carbonylation may serve as important biomarkers for numerous lung diseases, their role in PM_2.5_-induced lung inflammation has not been extensively investigated in human cells.

Many of the adverse health effects of PM_2.5_ exposure are hypothesized to be derived from oxidative stress, which is initiated by the formation of reactive oxygen species (ROS) within cells [24]. Although several studies have demonstrated the ROS potential and inflammatory action of diesel particulates from diesel engine combustion, the causative factors are less clear. Most studies use organics diluted and extracted from real exhaust particulates, and results reported may be skewed relative to cell exposure to the “complete” soot, i.e., PM_2.5_ along with its heteroatoms and condensed fractions. It is increasingly being recognized that extracts of diesel soot are a poor representation of the full range of toxicity and mutagenic effects [25]. Rather, the particles, and specifically their surface chemistry, are thought to have considerable direct health impacts. Yet, no studies have assessed and quantified the effect of parameters such as particle size, morphology, and surface chemistry in the cellular response to combustion-formed PM_2.5_ exposure. Therefore, we investigated these effects in human lung epithelial cells by using modified functionalized carbon blacks (soot) as surrogates for combustion-formed particulates. Our goal in using this approach was to understand the hierarchy of detrimental effects of soot from a combustion engine, subject to atmospheric aging, including the health effects of the primary particle itself, as compared to those of the functional groups on the carbonaceous backbone.

This work focuses on the identification of oxidative and pro-inflammatory factors induced as a response to exposure of human lung epithelial cells to synthetic soot in various surface chemistry forms. By exposing a male lung epithelial cell line (BEAS-2B) to lab-generated R-250 carbon black (nascent, nitric acid-treated, and ozone-treated carbon black), we tested the hypothesis that exposure to soot with different surface composition results in differential toxicity, inflammatory, and oxidative stress responses in human lung epithelial cells. We assessed the cell viability and expression of genes related to the inflammatory response and oxidative stress, including IL-1β, IL-6, superoxide dismutase 2 (SOD2), nuclear factor erythroid 2-related factor 2 (NFE2L2), and protein carbonylation levels at different time points and concentrations of soot. Our data indicate that the surface chemistry and concentration of soot, specifically higher oxygen content and concentration, play a critical role in the inflammatory and immune response initiated in the lung epithelium, and they are associated with a decrease in cell viability and an increase in protein carbonylation levels and expression of inflammatory cytokine genes.

## 2. Experimental Section

### 2.1. Synthesis and Characterization of Soot

Commercially produced carbon black (Regal 250, Cabot Corporation, Apharetta, GA, USA) was used as the model carbon black for its chemical purity and absence of organic content [25]. Synthetic soot was produced by functionalizing R250 carbon black via controlled oxidation by the methods described below. Four soot preparations were used for experiments:(1)Nascent soot (S1): Unmodified, non-oxidized, R250. This commercially produced carbon black was used as the model nascent soot form given its chemical purity and absence of organic content;(2)Nitric acid-treated soot (S2): Wet chemical treatment of R250 was conducted by treating 1 g of carbon black with 100 ml of laboratory-grade concentrated nitric acid (HNO_3_, >90%) at 80 °C under reflux for 24 h, just below the acid’s boiling point of 83 °C. The carbon–acid mixture was continuously stirred for uniform oxidation and functionalization. The mixture was maintained at a consistent simmer, and thereafter, it was washed with distilled water, filtered, and dried to obtain functionalized carbon black as synthetic soot;(3)Ozone-treated soot (S3): Dry gaseous treatment of carbon black was performed via exposure to ozone (O_3_) generated by subjecting oxygen (O_2_) to ultraviolet (UV) light. Ozone, a reactive gas, interacts with the carbon at room temperature and mildly oxidizes it, thereby functionalizing the carbon in the process. This method is a comparatively mild oxidative treatment than the wet acid reflux;(4)Nitric acid and heat-treated soot (S4): The powdered form of HNO_3_-treated carbon black was subjected to isothermal heat treatment at 300 °C in a hot-wall furnace for 1 h under an inert (Ar) environment.

All synthetic soot preparations were further characterized for their atomic oxygen content and functional groups introduced onto the carbon black surface. Characterization was performed by X-ray photoelectron spectroscopy (XPS) and thermogravimetric analysis (TGA), as indicated below.

### 2.2. X-Ray Photoelectron Spectroscopy (XPS)

X-ray photoelectron spectroscopy was used to identify and quantify possible different functional groups on the carbon surface and their contribution to the total surface atomic oxygen. XPS is a surface analysis technique based on the photoelectric effect where incoming X-rays of a known wavelength (λ) are used to eject surface electrons. With the known total energy of a photon, the kinetic energy (KE) of the ejected electron is measured, and the binding energy (BE) is calculated. The BE of a core–shell electron is characteristic of the element from which it is ejected and is used to identify the elements present. XPS experiments were performed using a Physical Electronics VersaProbe II instrument equipped with a monochromatic Al kα x-ray source (hν = 1486.7 eV) and a concentric hemispherical analyzer. Charge neutralization was performed using both low-energy electrons (<5 eV) and argon ions. The binding energy axis was calibrated using sputter-cleaned Cu foil (Cu 2p_3/2_ = 932.7 eV, Cu 2p_3/2_ = 75.1 eV). Peaks were charge referenced to the C-C band in the carbon 1s spectra at 284.5 eV. Quantification of the elements was done by curve-fitting of the high-resolution scan using the CasaXPS software.

### 2.3. Thermogravimetric Analysis (TGA)

A TA instruments Thermogravimetric Analyzer TA 5500 coupled with a Discovery Mass Spectrometer (MS) was used to analyze mass loss and the composition of the evolved gases. The temperature was ramped up at 5 °C/min in an inert atmosphere. Thermal analysis of the material gives a qualitative assessment of the volumetric uniformity of functionalization of the carbon; therefore, TGA is used as a bulk material characterization tool to complement the results observed by XPS. When subject to a steady temperature ramp, functional groups on the carbon oxidize (leave) at different temperatures. The subsequent mass loss curve can be used to qualitatively identify the oxygen functional groups present on the carbon blacks.

### 2.4. Cell Culture and Soot Exposure

Cells from the male bronchial epithelial cell line BEAS-2B (ATCC^®^ CRL-9609™) were thawed and cultured in RPMI-1640 medium (Gibco), supplemented with 2 mM glutamine (Gibco), 10% (*v*/*v*) heat-inactivated fetal bovine serum (VWR), and 10,000 units of Penicillin–Streptomycin (Gibco). Cells were grown in a 75 cm^2^ (T75) cell culture Flask (Corning) and incubated at 37 °C in a humidified chamber with 5% CO_2_.

#### 2.4.1. Cell and Soot Exposure for Cell Viability Assessment

For cell viability studies, BEAS-2B cells were plated in 24-well plates (Corning) at a density of 50,000 cells per well overnight prior to treatment with the soot preparations. The powder soot preparations were dissolved in a small amount of DMSO and then diluted in PBS. The final concentration of DMSO in the cell exposure medium was <1%. Moreover, the volume of soot solution added did not exceed 10% of the total media volume in the well. Cells were incubated with the soot preparations described above for either 6 h or 24 h at concentrations ranging from 1.56 µg/mL to 100 µg/mL at 37 °C in a humidified chamber with 5% CO_2_.

#### 2.4.2. Cell Culture and Soot Exposures for Gene Expression Assessment

To quantify the expression of inflammatory and oxidative stress genes, BEAS-2B cells were plated in 24-well plates (Corning) at a density of 50,000 cells per well overnight prior to treatment with the soot preparations. The powder soot preparations were dissolved in methanol (MeOH, negative control) and then diluted in PBS. The final concentration of MeOH in the cell exposure media was <0.5%. Moreover, the volume of soot solution added did not exceed 10% of the total media volume in the well. Cells were incubated for 6 h at 37 °C in a humidified chamber with 5% CO_2_ with all soot preparations, at concentrations of 1.56 to 12.5 µg/mL for inflammatory genes expression and of 3.125 µg/mL for oxidative stress genes expression.

#### 2.4.3. Cell Culture and Soot Exposures for Protein Carbonylation Assessment

For protein carbonylation studies, BEAS-2B cells were grown in a 75 cm^2^ (T75) cell culture Flask (Corning) and incubated at 37 °C in a humidified chamber with 5% CO_2_. To quantify protein carbonylation, cells were plated in 12-well plates (Corning) at a density of 100,000 cells per well overnight prior to treatment with the soot preparations. The powder soot preparations were dissolved in methanol (MeOH, negative control) and then diluted in PBS. The final concentration of MeOH in the cell exposure media was <1%. Moreover, the volume of soot solution added did not exceed 10% of the total media volume in the well. Cells were incubated for 24 h at 37 °C in a humidified chamber with 5% CO_2_ with all the soot preparations at concentrations of 25 µg/mL.

### 2.5. Cell Viability Assay

Cell viability was assessed using the MultiTox-Fluor Multiplex Cytotoxicity Assay Kit (Promega, Madison WI, USA), according to the manufacturer’s instructions at 6 h and 24 h post-exposure with the different soot preparations. The assay was tested for cell viability and cytotoxicity using specific positive controls: digitonin (toxicity) and ionomycin (necrosis).

### 2.6. RNA Purification and cDNA synthesis

After exposure to soot preparations, cells were harvested in TRizol (Life Technologies, Austin, TX, USA) and RNA was extracted using the Direct-zol kit (Zymo Research, Irvine, CA, USA) in the presence of DNAse. Total RNA was quantified by nanodrop measurement and stored at −80 °C until further analysis. For cDNA synthesis, 500 ng of RNA were retrotranscribed using the High Capacity cDNA kit (Life Technologies), following the manufacturer’s protocol. The cDNA reactions were stored at −20 °C until further use.

### 2.7. Real-Time PCR

The expression of inflammatory and oxidative stress-related genes was measured in 40 ng of cDNA by Real-Time PCR with TaqMan™ assays (Life Technologies). The following probes were used: IL1B (assay Hs01555410), IL6 (assay Hs00174131), NFE2L2 (assay Hs00232352), and SOD2 (assay Hs00167309). A housekeeping gene 18S (assay Hs03003631, Life Technologies) was used as a normalization control from 2 ng of cDNA. The reactions were conducted in triplicate using the TaqMan™ Fast Advanced Master Mix in 10 µL of final volume, following the manufacturer’s protocol. Expression results (Ct values) were monitored and extracted using the QuantStudio 12K Flex Software, and data were analyzed using the relative quantification method [26].

### 2.8. Total Protein Determination

After exposure to soot, cells were harvested in 150 µL of 20 mM Tris-HCl lysis buffer at pH 7.5. The total protein concentration in lysates was determined by the Bicinchoninic Acid (BCA) protein assay (Pierce), following the manufacturer’s protocol, using bovine serum albumin as a standard.

### 2.9. Protein Carbonylation Assay

The high sensitivity Protein Carbonyl ELISA Kit (Enzo Life Sciences, Farmingdale, NY, USA; cat #ALX-850-312-KI01) was used to determine the concentration of protein carbonylation starting from a sample volume containing 200 µg of protein and following the manufacturer’s protocol.

### 2.10. Statistical Analysis

Cell proliferation, gene expression, and protein carbonylation data are presented as means ± SE. Data were plotted, and statistical analyses were performed using the GraphPad Prism software (v.8.4.3). Interactions of treatment and concentration were determined by two-way ANOVA followed by Dunnett’s post hoc test, and differences among treatment groups were analyzed by one-way ANOVA followed by Tukey’s post hoc analysis with the GraphPad Prism software (v.8.4.3). Values of *p* < 0.05 were considered statistically significant.

## 3. Results

### 3.1. Characterization of Soot

An illustrative transmission electron micrograph (TEM) of a carbon black aggregate and primary particle is shown in Figure 1. After wet and dry chemical treatment, and prior to conducting cell exposures, the different soot preparations were characterized to determine their atomic oxygen content and functional groups introduced onto the carbon surface by XPS and TGA, as indicated below.

XPS was performed using survey and high-resolution scans, to identify the elements present on the surface of treated soot, specifically oxygen groups. As a baseline, nascent (untreated) carbon black (S1) was also subject to the same analytical procedure. Elements in each soot preparation were quantified via curve-fitting of the high-resolution scans using the commercial software CasaXPS. Table 1 shows the measured atomic percentages of the elements present in the analyzed soot. Table 2 gives the relative percentages of oxygen functional groups present in the soot samples. As expected, the oxygen content is higher in the acid-treated soot (S2) than the ozone-treated soot (S3) and acid + heat-treated soot (S4). Wet acid reflux treatment of carbon black resulted in ≈32 atomic % oxygen compared to a ≈10 atomic % from dry gaseous treatment, the latter being a relatively mild oxidant with oxidation performed at room temperature, explaining this difference in the type and degree of functionalization. Moreover, being the most reactive group, the -COOH functional group is selectively removed with heat treatment from the carbon black, significantly reducing the oxygen content to ≈12 at (%) overall as measured by XPS. TGA-MS revealed a negligible mass loss for the ozone-treated sample (S3) between 300 and 500 °C despite the curve fit value of ≈2 at (%) carboxylic assignment. Various factors account for this observation. First, lactone (i.e., ester) and carboxylic anhydride groups can also register as “carboxylic” groups by the XPS curve fit, given similar C1s binding energy. (Notably, these groups do not possess the labile hydrogen nor form an anionic state.) Possessing greater thermal stability, these groups do not decompose at 300 °C. Moreover, the 2 at (%) carboxylic content as extracted by deconvolution is near the minimal level detectable by this fitting procedure, which is estimated as ≈1 at (%). Correspondingly, ozone treatment is known to introduce minimal carboxylic (-COOH) functionality in carbons as well.

### 3.2. Cell Viability and Cytotoxicity in BEAS-2B Cells Exposed to Synthetic Soot

BEAS-2B cell viability was measured at 6 and 24 h after exposure to soot preparations (Figure 2 and Figure 3). A significant interaction of concentration and particle effect was found at both time points (*p* < 0.0001, two-way ANOVA). Treatment with ionomycin (positive control) resulted in a 95% reduction in cell viability after 6 h. When assessing independent effects, we found that after 6 h of treatment with S1 and S4 (soot with the lowest -COOH functional group content) (Table 2), there were no significant differences in cell viability at all the concentrations tested (Figure 2). Furthermore, the treatment of cells with soot containing the highest -COOH content (S2, S3) resulted in a reduction of viable cells in a dose-dependent manner. Treatment with S2 (9.4% -COOH content) resulted in a significant loss of cell viability at concentrations of 12.5 µg/mL and above. For S3 (2.4-COOH content), this effect was observed at a higher concentration (at least 50 µg/mL) (Figure 2).

At the 24-h time point, the exposure of cells to S1 and S4 at a concentration of 50 µg/mL or above induced a significant reduction in cell viability (*p* < 0.001) (Figure 3). The effect was observed at a much lower concentration (6.25 µg/mL and above) for S2, the soot with the highest -COOH content. Finally, the effect was observed at a concentration of 25 µg/mL or higher for S3 (Figure 3).

### 3.3. Expression of Pro-Inflammatory Genes in Cells Exposed to Synthetic Soot

To assess the ability of the different soot to induce an inflammatory response in BEAS-2B cells, we conducted real-time PCR experiments on extracts from cells exposed to S1–S4. To avoid significant cell death, we selected concentrations below 12.5 µg/mL and the 6-h time point based on the results obtained in Figure 2. We measured the expression of two pro-inflammatory cytokines (IL-1β and IL-6) that have been previously reported to increase in response to particulate matter exposure in human bronchial epithelial cells [20,27]. We limited our focus to IL-6 and IL-1β to compare the inflammatory effects of all soot preparations on IL-6 and IL-1β expression to these previously established outcomes. A significant interaction of concentration and particle effects was observed after 6 h of exposure to S1–S4 for both genes (IL-1β, *p* = 0.022; IL6, *p* < 0.0001, two-way ANOVA) (Figure 4 and Figure 5).

The expression of IL-1β was significantly increased in a dose-dependent manner, when cells where exposed to S2, S3, or S4 at concentrations above 3.125 µg/mL (Figure 4). At the lowest concentration tested (1.56 µg/mL), only soot with a higher total oxygen content and C=O groups (S2 and S4) induced significant changes in IL-1β expression (Figure 4). A comparison of IL-1β expression in cells exposed to all soot at the highest concentration (12.5 µg/mL) revealed a significant effect of functionalized particles (S2, S3, S4) vs. nascent particles (*p* < 0.05, one-way ANOVA) (Figure 6A). This effect was also observed at concentrations as low as 3.125 µg/mL (Figure 6B).

Similarly, the expression of IL-6 was significantly higher in cells exposed to functionalized soot (S2, S3, S4) at 6.25 µg/mL and 12.5 µg/mL, and it was significantly higher at 3.125 µg/mL only in cells exposed to S2 (Figure 5). When comparing the effects of different soot exposure at the highest concentration (12.5 µg/mL), we found significant differences among particles, with S2 inducing a significantly higher IL-6 mRNA expression than S3 and S4 (Figure 7A), and all three functionalized soot stimulating higher expression than the nascent (S1) particle (*p* < 0.05). In contrast, when comparing the effects of different soot at 3.125 µg/mL, the IL-6 response mimicked that of IL-1β (Figure 7B).

### 3.4. Expression of Oxidative Stress-Related Genes in Cells Exposed to Synthetic Soot

To assess the ability of the different soot preparations to induce the expression of oxidative stress-related genes in BEAS-2B cells, we conducted real-time PCR experiments on extracts from cells exposed to S1, S2, S3, and S4. We selected a concentration of 3.125 µg/mL at a 6-h time point considering that all soot preparations elicited changes in IL-6 and IL-1β gene expression at this concentration (Figure 4), while minimizing cell death (Figure 2). We measured the expression of two genes related to oxidative stress (SOD2 and NFE2L2), as previous studies have observed that PM_2.5_-induced oxidation alters the expression of these genes in respiratory tract cells [28]. Both NFE2L2 and SOD2 are critical molecules in the lung’s defense mechanism against oxidative stress, and their disruption enhances susceptibility to airway inflammation [29,30].

A significant interaction of particle effects on SOD2 and NFE2L2 expression (SOD2, *p* = <0.0001; NFE2L2, *p* = 0.0001, one-way ANOVA) was observed after 6 h of exposure to all soot particles at 3.125 µg/mL. SOD2 expression was significantly downregulated in cells exposed to S2, S3, and S4, but not S1 (Figure 8A). Cell exposure to S2 and S3 resulted in the most significant downregulation in SOD2 expression, both being the highest in O-C=O content compared to S1 and S4 (Table 2). Similarly, the expression of NFE2L2 was downregulated by cell exposure to all soot preparations, including S1, with the greatest effect observed in S2 and S3 exposure (Figure 8B). Comparison of SOD2 and NFE2L2 expression in cells exposed to all soot preparations at 3.125 µg/mL revealed a significant effect of functionalized particles (S1, S2, S3, S4) vs. control particles.

### 3.5. Protein Carbonylation in Cells Exposed to Synthetic Soot

To assess the ability of the soot preparations to alter protein carbonylation, we measured carbonylation levels in protein extracts from BEAS-2B cells exposed to S1, S2, S3, and S4 at a concentration of 25 µg/mL at a 24-h time point. We chose this concentration because it is the minimum concentration that results in a reduction in cell viability at this time point. The exposure of cells to S2 and S4, i.e., the soot with the highest oxygen, C-O, and C=O content, resulted in the greatest increase in total of protein carbonylation compared to S1 and S3 (Figure 9). Interestingly, the increase of protein carbonylation triggered by S2 and S4 mimics the effects observed on IL-1β expression at 12.5 µg/mL at 6 h of exposure.

## 4. Discussion

Air pollution is a significant health hazard that is associated with the worsening of cardiopulmonary disease and reduction of life expectancy. It is estimated that over 90% of the world population is exposed to higher than recommended levels of ambient air pollution, including outdoor particle pollution such as PM_2.5_ [31]. Decades of epidemiological and toxicological research have demonstrated that exposure to PM_2.5_ is detrimental to lung health and contributes significantly to the development and exacerbation of a multitude of lung diseases [32]. Nonetheless, the cellular and molecular mechanisms driving these outcomes remain poorly understood, and the influence of physicochemical characteristics of soot particles in these mechanisms has not been yet explored. Accumulating evidence from in vivo and in vitro studies using environmental samples or purified PM_2.5_ mixtures have indicated that the lung epithelium responds to particle exposures in a dose- and time-dependent manner, by activating redox responses, initiating and/or exacerbating inflammation, and causing DNA damage and epigenetic alterations [14]. In this study, rather than evaluating the effects of purified particles from environmental samples, we investigated the effects of exposing lung epithelial cells to synthetic soot. We assessed the effects of particles directly on oxidative stress and inflammatory responses to understand the contributions of their surface chemistry vs. the carbonaceous soot backbone itself. Our data revealed that surface chemistry, and specifically oxygen and carboxylic acid content, contributes to cell death, the expression of inflammatory and oxidative stress genes, and protein oxidative damage (carbonylation) in human lung epithelial cells in a concentration and time-dependent way.

Many of the adverse health effects of PM_2.5_ exposure are hypothesized to be derived from oxidative stress and inflammation, which is initiated by the formation of reactive oxygen species (ROS) and expression of pro-inflammatory genes within cells [33]. Although several studies have demonstrated the ROS potential and inflammatory action of diesel particulates from diesel engines [34], the causative factors are less clear. Most studies assessing these effects have used organics diluted and extracted from real exhaust particulates, and the results reported may be skewed relative to cell exposure to the “complete” soot, i.e., particulate matter along with its heteroatoms and condensed fractions. It is increasingly being recognized that extracts of diesel soot are a poor representation of the full range of toxicity [35]. Rather, the particles and, in particular, their surface chemistry have considerable direct impacts. To understand and potentially quantify the effect of parameters such as particle size, morphology, and chemistry, this study used surface-modified carbon black being a close substitute in its make and morphology. Functionalized carbon black is a surrogate for soot to understand the hierarchy of detrimental effects of soot from a combustion engine, i.e., the health effects of the primary particle itself as compared to those of the functional groups on the carbonaceous backbone. We chose carbon black to be able to control the extent of functionalization introduced and study the cell response by selectively modifying the material to exclude acidic functional groups. Exposure of a human lung epithelial cell line to model soot that closely resembles real combustion-generated soot in structure, particle size, and chemistry allowed us to identify and associate specific outcomes with particle physicochemical characteristics.

To our knowledge, this is the first study to examine the effects of surface particle chemistry and nanostructure on human lung epithelial responses. Prior studies using purified environmental samples have reported limited and often contradictory outcomes. Moreover, the majority of in vitro studies testing for toxicity use washing soot extracts that convolve condensed organics on the particles with extracted organics from the particle, ignoring the fixed particle surface chemistry. This has complicated the toxicity assessment of engine exhaust products, which is mainly due to our limited understanding of the specific contributions of different soot components. Our study provides new information on the contributions of surface chemistry to the particle’s cytotoxicity. Our data indicate that both oxygen content and functional groups enhance the particle’s ability to induce cell death. We demonstrate here that particles with the highest surface oxygen percentage and carboxylic acid content impair cell viability at lower concentrations than those with lower oxygen and carboxylic acid content. In addition, we show that this effect is also concentration- and time-dependent. Higher exposure times (24 h vs. 6 h) resulted in a significantly lower number of viable cells for a given concentration in particles with lower carboxylic acid content (S1, S4), but not in those with higher content (S2, S3).

It has been recently shown that exposure to PM_2.5_ below the current U.S. Environmental Protection Agency standards is associated with increased mortality [36]. This suggests that the inhalation of soot at very low concentrations may induce cellular damage that accumulates over time. While it is not possible to determine how closely the concentrations tested in this study reflect those of actual in vivo exposures in the human lung epithelium, we aimed to use concentrations that do not cause significant cytotoxicity when determining the contributions of particle chemistry on inflammation and oxidative stress effects. Thus, we examined the expression of two pro-inflammatory markers known to be expressed by lung epithelial cells (IL-1β and IL-6) [37] in BEAS-2B cells exposed for 6 h to soot at concentrations that were not toxic and did not have major effects on cell viability (12.5 µg/mL or lower). Our results demonstrate that the presence of functional groups in the particle’s surface was sufficient to induce an inflammatory response, as indicated by the stimulation of IL-1β and IL-6 gene expression by S2, S3, and S4, but not S1. Moreover, this effect was dose-dependent, indicating a potential mechanism involving the activation of pattern recognition receptors [38]. These receptors are known for recognizing specific ligands such as pathogen- and damage-associated molecular patterns, which are endogenous ligands derived from stressed cells [39]. The activation of pattern recognition receptors such as Toll-like-receptors in response to PM exposure results in the release of cytokines and chemokines to attract immune cells to the site of injury [40,41]. At the highest concentration tested (12.5 µg/mL), we did not observe a differential effect on IL-1β expression among different soot preparations. However, the expression of IL-6 mRNA was significantly higher in cells exposed to S2. This soot not only presents the higher oxygen atomic content, but also the higher percentage of -COOH, C=O, and C-O groups. In a previous study, 100 µg/mL ultrafine carbon black was observed to increase mRNA and protein expressions of IL-1β and IL-6 in primary rat epithelial lung cells, but IL-6 protein expression was delayed compared to IL-6 mRNA expression, and IL-1β mRNA and protein expression were not correlated [42]. Other studies have shown that IL-1β mRNA levels do not necessarily reflect IL-1β protein secretion [43]. Furthermore, it has been observed that endotoxin contamination, a component of some environmental PM preparations, may cause an intracellular accumulation of cytokines, including IL-1β [44], although previous findings indicate that endotoxin is usually found on coarse PM [45]. To further investigate the inflammatory effects of the synthetic soot, future studies should also examine the expression of IL-8 and TNF-α, as these genes have been implicated in numerous human lung diseases [46,47] and found to be upregulated in human bronchial epithelial cells exposed to PM [48,49].

In addition to studying the inflammatory response triggered by soot exposure, we examined the expression of SOD2 and NFE2L2, which play a critical role in antioxidant defense mechanisms against ROS. We used a concentration that did not significantly reduce cell viability (3.125 µg/mL) but was enough to elicit an inflammatory response by altering IL-6 and IL-1β gene expression. We found that the presence of functional groups in the particles’ surface was sufficient to alter the expression of both SOD2 and NFE2L2, as indicated by the gene expression changes in cells exposed to S1, S2, S3, and S4. Interestingly, SOD2 mRNA expression was significantly lower in cells exposed to all soot variants, including S1, but NFE2L2 gene expression was not significantly affected in cells treated with S1. The expression of SOD2 and NFE2L2 was the lowest in cells exposed to S2 and S3, the highest total oxygen carboxylic acid-containing soot, indicating a potential role of these groups in the inhibition of these gene’s expression.

Combined, these findings suggest that PM_2.5_ toxicity induces a decrease in the expression of antioxidant response genes SOD2 and NFE2L2. Under physiologic conditions, ROS generation is minimized by antioxidant proteins, but an overproduction of ROS disturbs the antioxidant defense system, leading to oxidative stress [30]. ROS are produced by the reduction of molecular oxygen and the formation of superoxide anions. Superoxides are precursors to most ROS but are metabolized by SOD2 to hydrogen peroxide (H_2_O_2_). Catalase (CAT) serves as a protective enzyme that breaks down H_2_O_2_ and eliminates ROS formation. Some studies have shown that exposure to PM_2.5_ induces a loss of SOD2 and CAT activity, leading to an accumulation of ROS [24]. PM_2.5_ has also been observed to downregulate protein kinase B (Akt) signaling, which is a known modulator of SOD2 and NFE2L2, decreasing the expression of SOD2 and NFE2L2 [49,50]. Another study observed that NFE2L2 expression in human BEAS-2B bronchial epithelial cells is downregulated when cells are subjected to high concentrations of PM_2.5_ or repeated exposure protocols [51].

An impairment in the antioxidant defense mechanism by the downregulation of SOD2 and NFE2L2 increases the expression of IL-1β and IL-6, inducing an inflammatory response, as observed in our findings [52]. Based on our data, a decrease in SOD2 and NFE2L2 gene expression is associated with a significant increase in IL-1β and IL-6, particularly in cells treated with S2 and S4, potentiating an inflammatory response as expected. The high oxygen content of S2 and S4 makes them potential good oxidants that can cause changes in oxygen saturation and can react with molecular oxygen to form H_2_O_2_ and the most potent form of ROS, hydroxyl radicals. Exposure to PM_2.5_ can downregulate CAT activity, causing an overaccumulation of these ROS, resulting in production and oxidative stress, increasing the inflammatory response [53]. While S3 (O_3_ treated PM_2.5_) did not cause the most significant increase in IL-1 β and IL-6 gene expression, S3 along with S2 did cause the most significant decrease in SOD2 and NFE2L2 gene expression. O_3_ generates ROS, including H_2_O_2_, thus exerting similar effects to soot with higher oxygen content [54]. Overall, more research is needed to elucidate the mechanisms by which the expression of antioxidant enzymes and transcription factors relates to oxidative stress responses induced by soot with varying surface chemistry.

Carbonylation is an irreversible protein modification induced by oxidative stress and is associated with chronic inflammation. Thus, we examined protein carbonylation as a potential biomarker of oxidative stress induced by exposure to PM_2.5_ in our studies. Protein carbonylation is a well-used biomarker for studying numerous human diseases, including Alzheimer’s disease, diabetes, and chronic lung disease. Despite this, very little is known about PM_2.5_-induced protein carbonylation and its role in oxidative stress. We exposed BEAS-2B cells for 24 h to soot preparations at a concentration that elicited changes in cell viability at 24 h (25 µg/mL) but no cell death. Protein carbonylation has been previously observed to increase in human keratinocytes exposed to PM_2.5_ at 50 µg/mL at a 24-h time point [23], indicating that changes in protein carbonylation require higher concentrations of PM than those required to elicit changes in gene expression. We found that that protein carbonylation was significantly higher in cells exposed to S2 and S4 compared to cells exposed to S1 and S3, thus following a similar trend observed in IL-1β and IL-6 gene expression. Targets of carbonylation are dependent on the oxidative environment of the cell and abundance of ROS. In the NFE2L2/Kelch-like ECH-associated protein 1 (KEAP1) mechanism, cytoplasmic KEAP1 binds NFE2L2, targeting it for proteasomal degradation. When carbonylated, KEAP1 releases NFE2L2, which translocates to the nucleus and, along with other transcription factors, binds to antioxidant response elements to produce cytoprotective proteins [22].

Our study has several limitations. First, while our characterization of lung epithelial cellular responses to synthetically modified soot revealed important contributions of surface chemistry, we used a submerged monolayer culture model wherein cells have been removed from their physiological conditions, including interactions with neighboring cells in the airway epithelium. Therefore, our studies will need to be replicated and investigated by alternative methods such as exposures in air–liquid interface cell cultures, using cell lines and primary cells of the lung and nasal epithelia, and using in vivo exposure models. Second, the BEAS-2B cell line is derived from an adult male individual; therefore, our results do not reflect the potential effects of sex/gender and/or age in the observed responses. Considering the known differential effects of air pollution exposure in men and women [55], and in individuals of different age groups and disease status [56], it will be important to expand these investigations to models that are representative of such populations. Third, while according to the literature, a particle concentration of 100 μg/mL corresponds to approximately 16μg/cm^2^ if all the suspended particles are deposited on the cells on the surface of the plates [57], we have not directly measured the proportion of particles that were in contact with cultured cells in our experiments. Finally, we have reported a differential reduction on the percent of viable cells as well as changes in mRNA expression of selected inflammatory and oxidative stress-related genes upon exposures, although in the current study, we have not evaluated changes in cytokine protein expression, the activity of oxidative stress enzymes, nor mechanisms previously associated with PM_2.5_ responses, including mitochondrial damage, apoptosis, autophagy, DNA damage, and epigenetic changes [52,58,59,60].

Our study has potential implications for environmental health. Vehicular traffic and consequent engine exhaust and industrial emissions are among the top contributors to outdoor pollution that results in millions of annual causalities worldwide. In light of the recent COVID-19 outbreak, one pilot (unpublished) study highlighted that an increase of only 1 μg/m^3^ in PM_2.5_ is associated with an 8% increase in the COVID-19 death rate [61]. While particle air pollution regulations are currently based on mass, the discussion is ongoing as to whether to include particle number concentration as a pseudo-surrogate for particle surface area. The results from our study reflect the impact of particle surface chemistry and physical structure as direct operative factors in health impacts, suggesting that such factors should be considered when developing environmental policies. These physicochemical properties are relatable to combustion conditions and fuel, indicating that precautions for exposure to small particles from various sources may be improved, and corresponding improvements in mitigation strategies may be more rapidly engineered.

In summary, we report that particle surface chemistry contributes to the cellular and molecular responses exerted in lung epithelial cells. We show that higher oxygen content and carboxylic acid functional groups in soot result in greater cell damage and death and cause the cells to react by activating inflammatory responses. These effects are exacerbated by soot concentration and exposure time. Together, our results demonstrate the role of particle surface chemistry as a direct operative factor impacting health. This physicochemical property is relatable to combustion conditions and fuel type. Thus, precautions for exposure to other soot types with similar surface (oxygen) chemistry may be improved, enabling better assessment from other anthropogenic sources and thereby formulating effective mitigation strategies to tackle this problem worldwide.

## 5. Conclusions

We conclude that particle surface chemistry, and specifically oxygen content and the presence of carboxylic acid groups in soot, differentially affect inflammatory and oxidative stress responses in human lung epithelial cells.

## Figures and Tables

**Figure 1 biomedicines-08-00345-f001:**
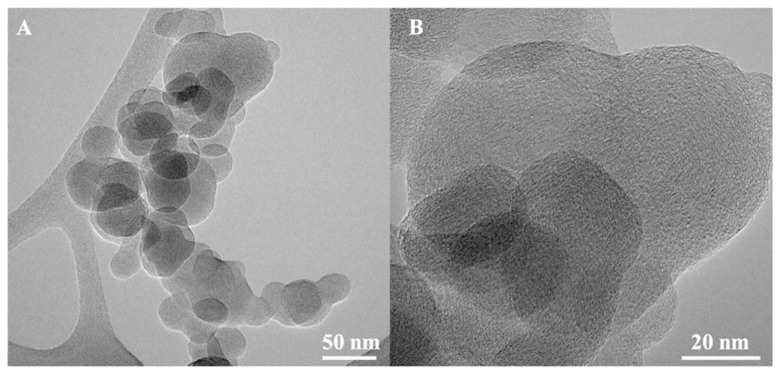
TEM image of a carbon black (**A**) aggregate and (**B**) primary particle.

**Figure 2 biomedicines-08-00345-f002:**
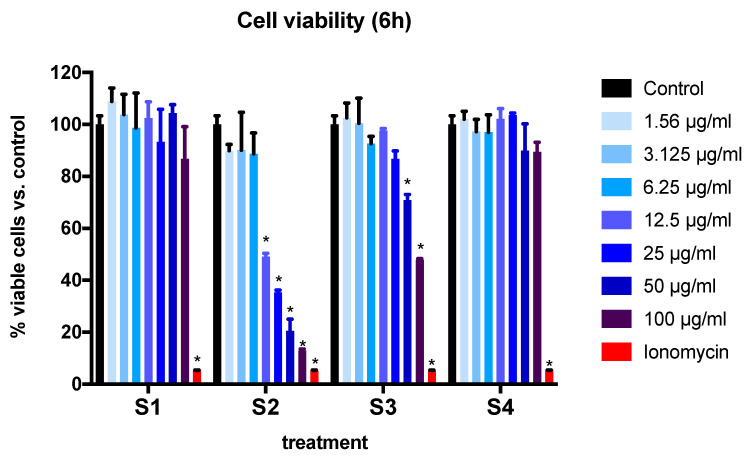
BEAS-2B cell viability expressed as percentage of viable cells at 6 h after exposure to four different soot preparations (S1–S4). The bars summarize data from three independent experiments (*n* = 3 replicates per experiment) with results normalized to control (cells exposed to DMSO). Data are expressed as mean ± SEM. * *p* < 0.001 (Dunnett’s post hoc multiple comparisons test). Two-way ANOVA interaction: *p* < 0.0001, F (21, 40) = 7.640; concentration effect: *p* < 0.0001, F (7, 40) = 25.67); soot-type effect (*p* < 0.0001, F (3, 40) = 69.91. For a description of S1–S4, see Section 2.

**Figure 3 biomedicines-08-00345-f003:**
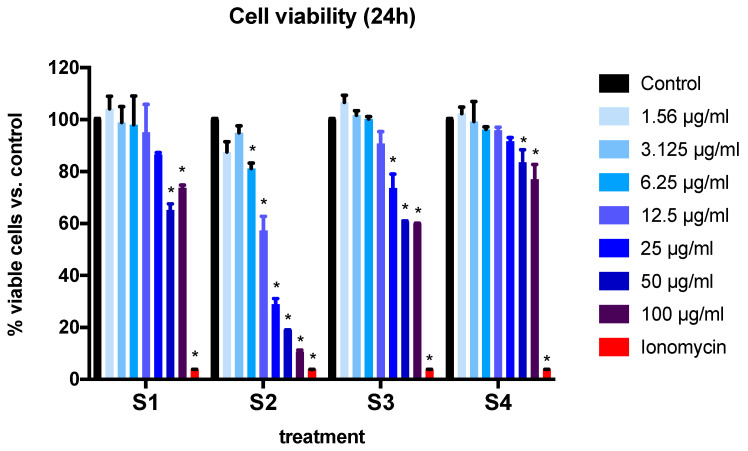
BEAS-2B cell viability expressed as percentage of viable cells after exposure to four different soot preparations for 24 h. The graph summarizes data from three independent experiments (*n* = 3 replicates per experiment), with results normalized to control cells (exposed to DMSO). Data are expressed as mean ± SEM. * *p* < 0.001 (Dunnett’s post hoc multiple comparisons test). Two-way ANOVA interaction: *p* < 0.0001, F (21, 36) = 11.59; concentration effect: *p* < 0.0001, F (7, 36) = 100.4; soot type effect (*p* < 0.0001, F (3, 36) = 127.6). For a description of S1–S4, see Section 2.

**Figure 4 biomedicines-08-00345-f004:**
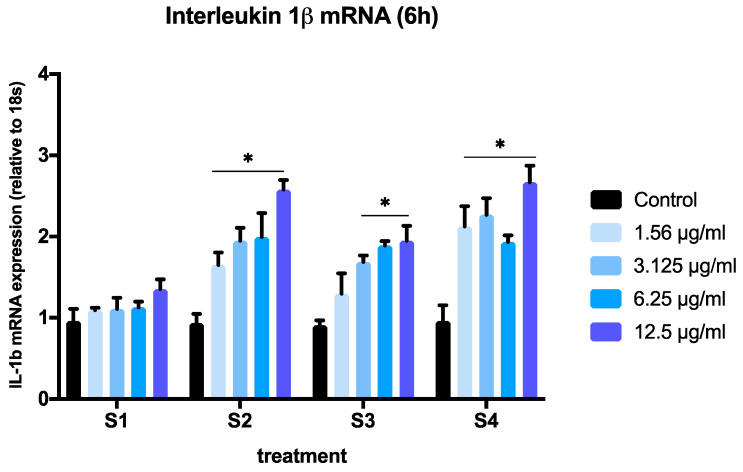
Expression of human interleukin-1 mRNA (IL1B) (relative to 18S expression) in BEAS-2B cell after exposure to soot preparations for 6 h. The graph summarizes data from three independent experiments (*n* = 3 replicates per experiment), with results normalized to controls (exposed to methanol) and expressed as mean ± SEM. Two-way ANOVA interaction: *p* = 0.0109, F (12, 71) = 2.415; concentration effect: *p* < 0.0001, F (4, 71) = 30.51; soot type effect (*p* < 0.0001, F (3, 71) = 15.48). * *p* < 0.05, different from control (Dunnett’s multiple comparison’s test). For a description of S1–S4, see Section 2.

**Figure 5 biomedicines-08-00345-f005:**
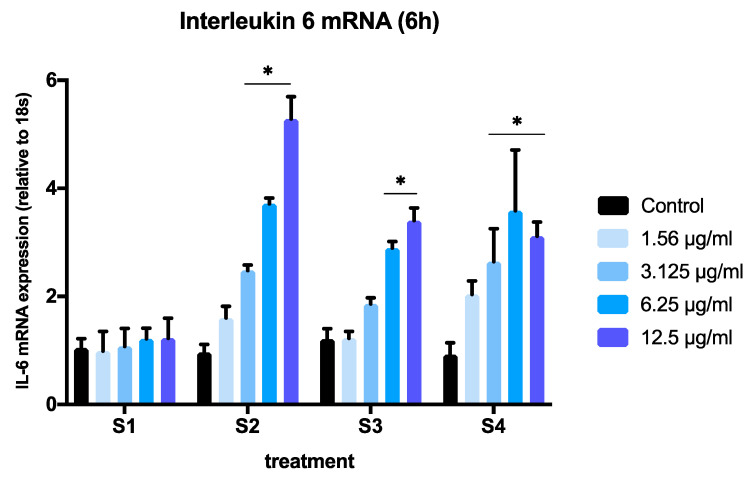
Expression of human interleukin-6 mRNA (IL6) (relative to 18S expression) in BEAS-2B cell after exposure to soot preparations for 6 h. The graph summarizes data from three independent experiments (*n* = 3 replicates per experiment), with results normalized to controls (exposed to methanol) and expressed as mean ± SEM. Two-way ANOVA interaction: *p* < 0.0001, F (12, 61) = 5.563 concentration effect: *p* < 0.0001, F (4, 61) = 36.99; soot type effect (*p* < 0.0001, F (3, 61) = 22.84). * *p* < 0.05, different from control (Dunnett’s multiple comparison’s test). For a description of S1–S4, see Section 2.

**Figure 6 biomedicines-08-00345-f006:**
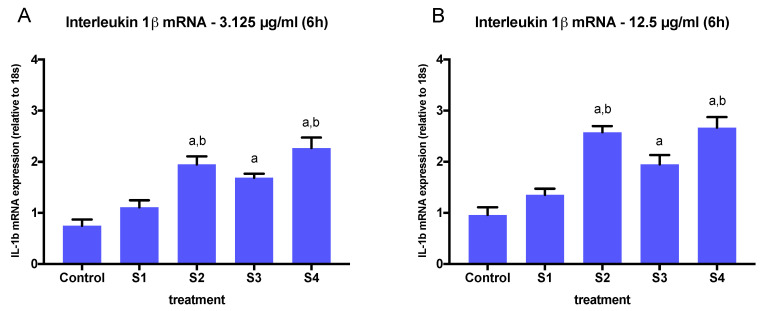
Human IL-1β mRNA (IL1B) expression (relative to 18S expression) in BEAS-2B cells treated with S1–S4 soot at 3.125 µg/mL (**A**) or 12.5 µg/mL (**B**) for 6 h. The graphs summarize data from three independent experiments (*n* = 3 replicates per experiment), with results normalized to controls (exposed to methanol) and expressed as mean ± SEM. a: different from control, b: different from S1 (*p* < 0.05). One-way ANOVA (*p* < 0.001, F (4, 20) = 20.47).

**Figure 7 biomedicines-08-00345-f007:**
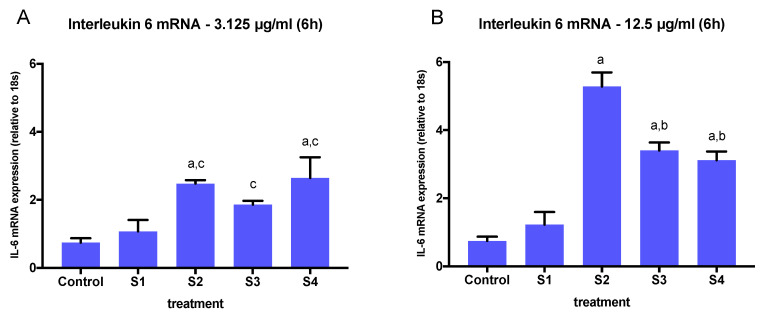
Human IL-6 mRNA (IL6) expression (relative to 18s) in BEAS-2B cells treated with S1–S4 soot at 3.125 µg/mL (**A**) or 12.5 µg/mL (**B**) for 6 h. The graphs summarize data from three independent experiments (*n* = 3 replicates per experiment), with results normalized to controls (exposed to methanol) and expressed as mean ± SEM. a: different from control and S1 (*p* < 0.001), b: different from S2 (*p* < 0.05), c: different from control. One-way ANOVA (*p* < 0.001, F (3, 8) = 25.7).

**Figure 8 biomedicines-08-00345-f008:**
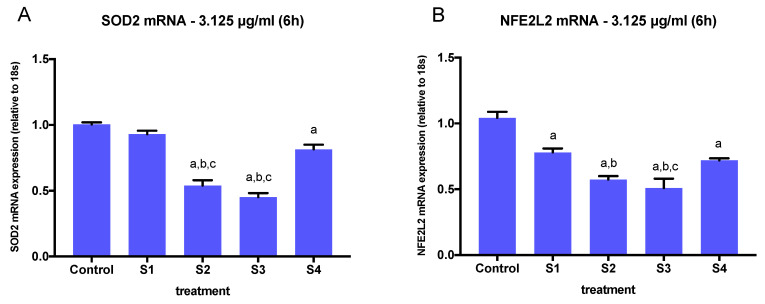
Expression of oxidative stress genes (relative to 18S expression) in BEAS-2B cells treated with S1–S4 soot at 3.125 ug/mL for 6 h. (**A**) Superoxide dismutase 2 (SOD2) mRNA. (**B**) Nuclear factor erythroid 2-related factor 2 (NFE2L2) mRNA. The graphs summarize data from three independent replicates, with gene expression results normalized to controls (exposed to DMSO) and expressed as mean ± SEM. a: different from control, b: different from S1, c: different from S4. One-way ANOVA (*p* < 0.05).

**Figure 9 biomedicines-08-00345-f009:**
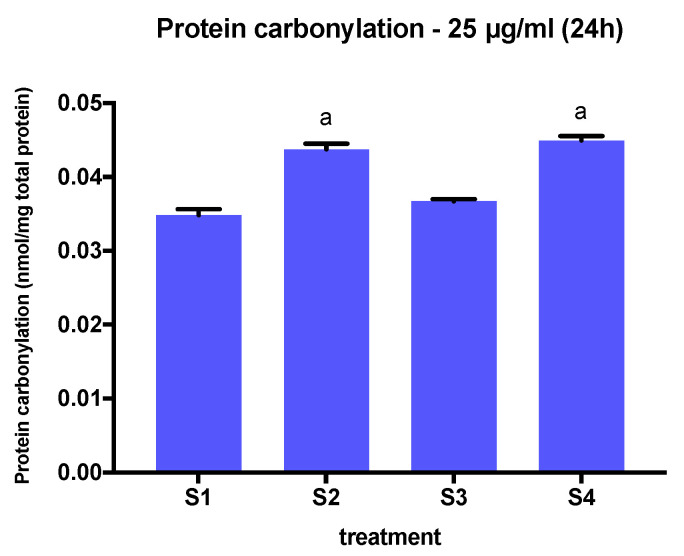
Protein carbonylation levels in BEAS-2B cells treated with S1–S4 soot at 25 ug/mL for 24 h. Results are expressed as nmol/mg of total protein. a: different from S1 and S3. One-way ANOVA (*p* < 0.001, F (3, 12) = 63.07).

**Table 1 biomedicines-08-00345-t001:** Elemental content measured as atomic percent for samples S1–S4. S1: nascent soot, S2: nitric acid-treated soot, S3: ozone-treated soot, S4: nitric acid and heat-treated soot.

Soot	Treatment	Measured Atomic ^1^ %			
		C	O	N	S
S1	None	97.2	1.3	--	0.9
S2	HNO_3_	67.8	31.5	1.3	--
S3	Ozone	90	9.4	--	0.6
S4	HNO_3_ + 300 °C	86.5	13.2	--	0.2

^1^ C: carbon, O: oxygen, N: nitrogen, S: sulfur.

**Table 2 biomedicines-08-00345-t002:** Oxygen group percentages for samples S1–S4.

Soot	Treatment	Oxygen Groups %			
		C-O	C=O	O-C=O	Total O
S1	None	--	--	--	1.3
S2	HNO_3_	10.2	4.9	9.4	34.0
S3	Ozone	2.2	1.4	2.4	8.3
S4	HNO_3_ + 300 °C	7.3	3.8	0.5	12.1

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
