# Peer review of "Identification of Toxicity Parameters Associated with Combustion Produced Soot Surface Chemistry and Particle Structure by in Vitro Assays"

_biomedicines, 2020, doi:10.3390/biomedicines8090345_

Round 1
Reviewer 1 Report
Article title: Identification of toxicity parameters associated with combustion produced soot surface chemistry and particle structure by in vitro assays
This manuscript describes a series of focused in vitro experiments assessing the contribution of surface chemistry of soot particles on inflammatory and oxidative stress responses in human bronchial epithelial cells (BEAS-2B). Lab generated soot particles with different surface chemistries are used as surrogates for PM2.5 found in ambient air pollution. Although this study was well executed, there are several weaknesses related to the methodology and the choice of exposure model, which significantly limits any relevancy to real-world human exposure scenarios.
There is no mention of any characterization of the particles in the media that was used to treat the cells. How do the authors know whether DMSO or methanol (solvents that are used to prepare the soot particles solution to which the cells are exposed) do not induce any changes in the surface chemistry of the soot particles? What are the physical and chemical characteristics of the particles in the media that is used for the cellular exposure? What are the hydrodynamic diameters of the soot particles when in solution? Any measurements made using dynamic light scattering (DLS) for example? Do the different surface chemistries of the particles impact on how they agglomerate when in solution?
Introduction: It is unclear what the authors are trying to say: if air pollution was responsible for 7 million deaths annually worldwide, and contributed to 5 million premature deaths in 2017… is the situation improving?
Introduction: L54-56. As there is nothing in this study that refers to COVID-19, I would remove this sentence from the introduction.
Methods: For the gene expression: why expose the cells to different concentrations of the soot particles dissolved in DMSO to evaluate genes related to inflammation and genes related to oxidative stress?
For the evaluation of the proteins, why were the soot particles dissolved in methanol, while for the other assays (e.g. gene expression) the particles were dissolved in DMSO?
Why were the cells exposed to higher concentrations of soot particles for the protein assessment compared to the concentrations used for the gene expression analysis?
All these differences (solvent used, exposure concentrations) for all assays must be scientifically justified and explicitly stated in the manuscript.
Results: The data is presented as 3 independent replicates, which is good; however, for scientific rigor each experiment consisting on a n = 3 (independent replicates) must be repeated at least 3 independent times. Please clarify and specify how scientific rigor was addressed in this study.
For the gene inflammatory response, only 2 genes were evaluated: IL-1b and IL-6. Please justify why only those 2 genes were selected.
For the oxidative stress gene response, only 2 genes were evaluated: SOD2 and NFE2L2. Please justify why only those 2 genes were selected.
Discussion: L406-407: This sentence is an overstatement. Publications in the field of nanotoxicology have already address the effects of similar variables on fine and ultrafine (nano) particles. Please remove this sentence.
L518-521: Overstatement. The model used in this study is inadequate to extrapolate the results to any policy. As mentioned by the authors in the limitations section, they used cell cultures, which were exposed to particles in a suspension media. This is not a physiologically relevant exposure models, as air-liquid interface (ALI) in vitro exposure models are much better suited for this kind of experiments.
Author Response
This manuscript describes a series of focused in vitro experiments assessing the contribution of surface chemistry of soot particles on inflammatory and oxidative stress responses in human bronchial epithelial cells (BEAS-2B). Lab generated soot particles with different surface chemistries are used as surrogates for PM2.5 found in ambient air pollution. Although this study was well executed, there are several weaknesses related to the methodology and the choice of exposure model, which significantly limits any relevancy to real-world human exposure scenarios.
Response: We thank the reviewer for their comments. We have considered all their suggestions and observations and we have modified the manuscript accordingly (please see attached new version).
There is no mention of any characterization of the particles in the media that was used to treat the cells. How do the authors know whether DMSO or methanol (solvents that are used to prepare the soot particles solution to which the cells are exposed) do not induce any changes in the surface chemistry of the soot particles? What are the physical and chemical characteristics of the particles in the media that is used for the cellular exposure? What are the hydrodynamic diameters of the soot particles when in solution? Any measurements made using dynamic light scattering (DLS) for example? Do the different surface chemistries of the particles impact on how they agglomerate when in solution?
Response: The oxygen groups introduced to the soot are stable in these organic media. We note that carbohydrates, glyco-proteins, and related cellular biomolecules bear these same functional groups. The reagents will not alter the carbon surface, far harsher reaction conditions are necessary - please see manuscript re: nitric acid treatment (reflux) required to introduce these oxygen groups. Carbon is an inert material. It is to be expected that some water (molecules) of hydration will surround the particles but this molecular layer (10’s Angstroms) will be negligible relative to the particle size of 10’s nanometers and aggregate dimensions of a few hundred nanometers. The aggregate structure and primary size are reported in the text, and valid for their dispersion in solution, as is common for other carbon nanomaterials similarly modified. The “like” groups and similar surface charge upon the modified carbon black particles provide dispersion by steric and electrostatic repulsion forces, as evidenced by the suspension stability w/o flocculation or settling behavior. The hydrodynamic size would logically be somewhat smaller than the geometric outline of the fractal particles, but is inconsequential for these studies as the “aggregate diameter” of order 100 nm is far less than the few micron cellular size. Particles clearly will follow induced flow in stirring/shaking, but otherwise undergo Brownian motion. Thus to summarize, the physical characteristics of the particles in the media to which the cells are exposed are those represented by the micrograph and reported size characteristics. The chemical form of the particles in the media is that reported by XPS analyses of total surface oxygen and distribution of oxygen functional groups. DLS is impeded by the highly absorbing nature of the particles. We note that commercial carbon inks have traditionally been based on carbon black suspensions with similar dispersion formulation and stability.
Introduction: It is unclear what the authors are trying to say: if air pollution was responsible for 7 million deaths annually worldwide, and contributed to 5 million premature deaths in 2017… is the situation improving?
Response: These are singular statistics to convey the severity of the problem, but not meant to convey trend. We have now revised the manuscript’s abstract to avoid confusion.
Introduction: L54-56. As there is nothing in this study that refers to COVID-19, I would remove this sentence from the introduction.
Response: As more studies are published highlighting the impact of air pollution on COVID-19 outcomes, especially lung inflammation, we believe that this statement is relevant to our work and highlights the importance of characterizing health effects of particle pollution exposure. We have now moved this sentence to the discussion (page 15, line 536)
Methods: For the gene expression: why expose the cells to different concentrations of the soot particles dissolved in DMSO to evaluate genes related to inflammation and genes related to oxidative stress?
Response: Prior studies have shown that the response to PM exposure is concentration-dependent. Therefore, we selected a range of concentrations to evaluate differences in soots with varying surface chemistry.
For the evaluation of the proteins, why were the soot particles dissolved in methanol, while for the other assays (e.g. gene expression) the particles were dissolved in DMSO?
Response: Particles were dissolved in methanol for both experiments (gene expression and protein carbonylation). We have corrected the methods section in the revised version.
Why were the cells exposed to higher concentrations of soot particles for the protein assessment compared to the concentrations used for the gene expression analysis?
Response: The available literature indicates that changes in protein carbonylation require higher concentrations of particulate matter than those needed to observe changes in gene expression. For the purposes of our pilot study assessing differences among soot preparations, we chose a concentration that had been reported to exert changes in protein carbonylation in epithelial cells. We provide additional justification for this approach in the revised version of the manuscript (page 11, line 364).
All these differences (solvent used, exposure concentrations) for all assays must be scientifically justified and explicitly stated in the manuscript.
Response: We have now revised the manuscript to further justify these approaches.
Results: The data is presented as 3 independent replicates, which is good; however, for scientific rigor each experiment consisting on a n = 3 (independent replicates) must be repeated at least 3 independent times. Please clarify and specify how scientific rigor was addressed in this study.
Response: Studies were repeated at least 3 times with n=3 replicates for all experiments (except protein carbonylation). We have clarified this in the revised version of the paper.
For the gene inflammatory response, only 2 genes were evaluated: IL-1b and IL-6. Please justify why only those 2 genes were selected.
Response: For the purpose of our study (i.e. comparing the effects of different synthetic soot preparations on previously described outcomes) we limited our focus to these two genes because of the large body of literature indicating their increased expression in response to particulate matter exposure. We have now provided a justification for this approach in the revised manuscript (page 8, line 289)
For the oxidative stress gene response, only 2 genes were evaluated: SOD2 and NFE2L2. Please justify why only those 2 genes were selected.
Response: We limited our focus on these two genes as they are critical molecules in the oxidative stress response in the lung. We now provide additional justification for this approach in the revised manuscript (page 10, line 341).
Discussion: L406-407: This sentence is an overstatement. Publications in the field of nanotoxicology have already address the effects of similar variables on fine and ultrafine (nano) particles. Please remove this sentence.
Response: While the reviewer is correct in that other studies have assessed the effects of purified nanoparticles from ambient samples, ours is the first study investigating the effects of specific functional groups in the particle surface by exposing cells to characterized synthetic soot.
L518-521: Overstatement. The model used in this study is inadequate to extrapolate the results to any policy. As mentioned by the authors in the limitations section, they used cell cultures, which were exposed to particles in a suspension media. This is not a physiologically relevant exposure models, as air-liquid interface (ALI) in vitro exposure models are much better suited for this kind of experiments.
Response: We have now edited this statement to clarify its meaning.
Reviewer 2 Report
In this manuscript Housseiny et al. analyze the association between toxicity and the surface particle chemistry of carbon black (CB) particles (as a model of soot-like PM air pollution) on the human lung epithelial cell line BEAS-2B. To this end, authors expose BEAS-2B cells 6h or 24h to CB that have been intentionally modified under laboratory conditions to generate particles with different content in oxygen groups, particularly in carboxyl groups. After exposure to modified CB at different concentrations, authors measure cytotoxicity, induction of inflammation (mRNA expression of IL-1b and IL-6) and oxidative stress/damage (mRNA expression of superoxide dismutase 2 and nuclear factor erythroid 2-related factor 2 and protein carbonylation). Authors observe that higher content in carboxyl groups positively correlates with toxicity and proinflammatory activity associated to CB exposure. Although data are interesting in the field of environmental toxicology, conclusions are supported by preliminary data that must be completed.
Major points
-Fold changes at mRNA levels in IL-1b and IL-6 expression are quite modest. Authors should measure protein levels in supernatants or cell lysates, especially for IL-1b a cytokine which upregulation at mRNA does not necessary correlate with the secretion of the mature protein (inflammasome-dependent). Further, CXCL-8 (IL-8) cytokine also rapidly upregulated after exposure of lung epithelial cells to PM (Ref 21), authors could measure IL-8 in supernatants.
- Do Fig 6 (IL-1b mRNA) and Fig 7 (IL-6 mRNA) provide new data comparing with Fig 4 and Fig 5, respectively? Seems that Fig 6 and Fig 7 represent the same data/information that Fig 4 and Fig 5, therefore Fig 6 and 7 could be removed.
-To quantify oxidative stress authors only measure the mRNA levels of SOD2, NFE2L2 (antioxidant molecules) and protein carbonylation (protein oxidative damage). Here again, effects of CB are quite modest even for those more active (S2, S3) (Fig-8, Fig-9). Authors should include positive controls to evaluate the extend of these changes. In addition, authors should measure the formation of ROS directly to confirm the induction and the extend of oxidative stress by the modified carbon black particles tested.
Minor points
- Fig 2 and 3 could be merged in one figure to better compare effects on cytotoxicity at 6h and 24h
-Authors should measure endotoxin(LPS) content in unmodified as well as modified CB.
Author Response
In this manuscript Housseiny et al. analyze the association between toxicity and the surface particle chemistry of carbon black (CB) particles (as a model of soot-like PM air pollution) on the human lung epithelial cell line BEAS-2B. To this end, authors expose BEAS-2B cells 6h or 24h to CB that have been intentionally modified under laboratory conditions to generate particles with different content in oxygen groups, particularly in carboxyl groups. After exposure to modified CB at different concentrations, authors measure cytotoxicity, induction of inflammation (mRNA expression of IL-1b and IL-6) and oxidative stress/damage (mRNA expression of superoxide dismutase 2 and nuclear factor erythroid 2-related factor 2 and protein carbonylation). Authors observe that higher content in carboxyl groups positively correlates with toxicity and proinflammatory activity associated to CB exposure. Although data are interesting in the field of environmental toxicology, conclusions are supported by preliminary data that must be completed.
Response: We thank the reviewer for their comments. We have considered all their suggestions and observations and we have modified the manuscript accordingly (please see attached new version).
Major points
-Fold changes at mRNA levels in IL-1b and IL-6 expression are quite modest. Authors should measure protein levels in supernatants or cell lysates, especially for IL-1b a cytokine which upregulation at mRNA does not necessary correlate with the secretion of the mature protein (inflammasome-dependent). Further, CXCL-8 (IL-8) cytokine also rapidly upregulated after exposure of lung epithelial cells to PM (Ref 21), authors could measure IL-8 in supernatants.
Response: We thank the reviewer for the suggestion. While we have limited the measured outcomes to transcriptional effects in this pilot paper comparing different soot preparations, we now discuss additional implications and future directions in more detail in the revised manuscript (page 13, line 456).
- Do Fig 6 (IL-1b mRNA) and Fig 7 (IL-6 mRNA) provide new data comparing with Fig 4 and Fig 5, respectively? Seems that Fig 6 and Fig 7 represent the same data/information that Fig 4 and Fig 5, therefore Fig 6 and 7 could be removed.
Response: Figures 4 and 5 represent concentration-response histograms. These are important as prior work has shown that the effects of PM exposure are concentration-dependent. In Figures 6 and 7, we compare the different soot preparations (i.e. the objective of our study) at selected concentrations. We believe it is important to show why these concentrations were selected by showing Figures 4 and 5, but we are open to move these to a supplementary material section if the reviewer and editor consider it pertinent.
-To quantify oxidative stress authors only measure the mRNA levels of SOD2, NFE2L2 (antioxidant molecules) and protein carbonylation (protein oxidative damage). Here again, effects of CB are quite modest even for those more active (S2, S3) (Fig-8, Fig-9). Authors should include positive controls to evaluate the extend of these changes. In addition, authors should measure the formation of ROS directly to confirm the induction and the extend of oxidative stress by the modified carbon black particles tested.
Response: We agree with the reviewer in that these studies should be conducted in the future. Our current study focused on previously described transcriptional effects of inflammatory and oxidative stress genes as primary outcomes to compare the effects of soot with varying surface chemistry. We have now expanded the discussion section to include additional future directions for this pilot study.
Minor points
- Fig 2 and 3 could be merged in one figure to better compare effects on cytotoxicity at 6h and 24h
Response: We thank the reviewer for the recommendation. Because of the size of these figures and the number of bars in each histogram, merging the two figures in one is not possible without compromising the size of the fonts.
-Authors should measure endotoxin(LPS) content in unmodified as well as modified CB.
Response: We thank the reviewer for the suggestion. We agree with the reviewer in that endotoxin contamination may cause intracellular accumulation of cytokines, therefore we have acknowledged this possibility in the revised manuscript (page 13, line 453).
Round 2
Reviewer 1 Report
Most of the comments were addressed adequately.
Reviewer 2 Report
Authors have partially responded to reviewer's comments